# First Experiences with Newborn Screening for Congenital Hypothyroidism in Ulaanbaatar, Mongolia

**DOI:** 10.3390/ijns7020029

**Published:** 2021-06-07

**Authors:** Altantuya Tsevgee, Khishigjargal Batjargal, Tsolmon Munkhchuluun, Naranchimeg Khurelbaatar, Gerelmaa Nansal, Oyun-Erdene Bulgan, Sumberzul Nyamjav, Gerelmaa Zagd, Erdenetuya Ganbaatar

**Affiliations:** 1Department of Pediatrics, School of Medicine, Mongolian National University of Medical Sciences, Ulaanbaatar 14210, Mongolia; ts.altantuya1127@gmail.com (A.T.); khishigjargal0802@gmail.com (K.B.); gerelmaa@mnums.edu.mn (G.Z.); 2Department of Pediatrics, Jichi Medical University, 3311-1 Yakushiji, Shimotsuke-shi, Tochigi 329-0498, Japan; 3Department of Health Promotion and Disease Prevention, National Center for Public Health, Ulaanbaatar 13381, Mongolia; tsolmon01@gmail.com; 4Children’s Hospital, National Center for Maternal and Child Health, Bayangol District, Ulaanbaatar 16060, Mongolia; kh_naraa08@yahoo.com (N.K.); Ngerelmaa@yahoo.com (G.N.); 5Screening Diagnostic Reference Center, Bayangol District, Ulaanbaatar 16091, Mongolia; oyuda_b@yahoo.com; 6Graduate School, Mongolian National University of Medical Sciences, Ulaanbaatar 14210, Mongolia; sumberzul@mnums.edu.mn

**Keywords:** congenital hypothyroidism, neonatal screening, blood spot test, filter paper

## Abstract

Congenital hypothyroidism (CH) is among the most common conditions leading to intellectual disability, which can be prevented by early detection through newborn screening (NBS). In Mongolia, a regional screening program for CH was launched in 2000, which was supported by the International Atomic Energy Agency (IAEA) for the Asia Pacific Region. In our present study, a total of 23,002 newborns from nine districts in Ulaanbaatar were screened between 2012 and 2020, by the measurement of the thyroid-stimulating hormone (TSH) from dried blood spots, sampled 24 to 72 h after birth. The level of TSH was measured by the DELFIA assay. The overall CH prevalence confirmed at birth was 1/2091. The female-to-male ratio for CH cases was 1.8:1. The majority of patients were asymptomatic (72.7% of CH cases); umbilical hernia and cold or mottled skin were reported symptoms in patients with CH (27.3%). Thyroid dysgenesis (hypoplasia and agenesis) was the most common etiology, with a total of nine cases (81.8%) out of the eleven patients. The lapse between the birth date and the initiation of L-thyroxine treatment in CH-positive children was lower than 15 days in 63.64% of cases or 15 to 30 days in 36.36% of children. Further research is required to expand the screening coverage for CH in Mongolia.

## 1. Introduction

Congenital hypothyroidism (CH) is among the most common neonatal endocrine disorders and the few preventable causes of severe neurological and psychiatric impairment CH affects approximately one in 3000 to 4000 live births worldwide [1,2]. Recent reports suggest that the prevalence of CH is increasing; in particular, the prevalence of CH among Asian infants is higher than other ethnicities [3,4]. Most neonates born with CH have normal appearance and no detectable physical signs. Asymptomatic CH results in severe neurodevelopmental impairment if treatment is delayed [5]. Therefore, NBS is mandatory to screen neonates for CH for early diagnosis and treatment.

Over the last 40 years, NBS programs have been established in most developed countries, measuring thyroid-stimulating hormone (TSH) and thyroxine (T4) in dried blood spots, and have allowed the early detection of CH [6]. However, NBS programs have not been able to be implemented in developing countries, including Mongolia. There is increased awareness of the frequency and importance of severe neurodevelopmental deficits that result from delayed diagnosis [7]. Among the Mongolian population, the prevalence and clinical forms of CH are studied insufficiently. In the past two decades, only one study conducted in Mongolia found the prevalence of CH of 1 in 1892 live births [8]. There have been no other studies that have investigated the prevalence rate and clinical forms of CH in Mongolia since 2002.

The aim of this study was to assess the prevalence of CH in Ulaanbaatar, Mongolia between 2012 and 2020.

## 2. Materials and Methods

### 2.1. Patient Population

From 2012 to 2020, a total of 23,002 neonates were screened in Ulaanbaatar for CH. During this period, a total of nine districts joined the screening program, involving a maximum of six maternity hospitals (out of seven in Ulaanbaatar) (Figure 1). Based on the goal, one maternity hospital was excluded from our study due to referred mothers from provinces. The suspected CH neonates were recalled, confirming CH. CH was diagnosed with a report of thyroid ultrasound and the results of laboratory tests. Ultrasonography of the thyroid gland was performed during the first month of life.

### 2.2. Newborn Screening for CH

The newborn screening for CH was based on the measurement of thyroid-stimulating hormone (TSH) in dried blood spots on filter paper specimens. Dried blood samples on filter paper were collected from newborns born in Ulaanbaatar at 24 to 72 h of age from public and private maternity hospitals by the Screening Diagnostic Reference Center. According to the guidance of NBS developing programmes from IAEA [9] and the Centers for Disease Control and Prevention (CDC) [10], we evaluated valid or invalid blood specimens on filter paper before screening. Thus, inadequately collected blood samples or invalid blood specimens for TSH determination were excluded and new samples were collected by trained nurses at the maternity hospitals. A time-resolved fluoroimmunoassay was used to measure the TSH concentration in the dry blood spot specimen using a DELFIA assay (AutoDELFIA kit of neonatal human TSH dry blood, Perkin Elmer, Turku, Finland). A cut-off value of 20 mIU/L was used for the whole blood TSH in dried blood spots during this period. Since a range of cut-offs from 10 to 30 mIU/L were used by different programs and countries [11,12,13], a trial was conducted from 2012 to 2015 to define the most efficient threshold for our conditions. When the TSH value on the filter paper was <20 mIU/L, it was considered negative, and no further action was pursued. TSH values ≥ 20 mIU/L on the filter paper were considered positive for CH, and a new dried blood specimen was requested for the second screening. In the second screening, if the TSH values were still ≥16 mIU/L in dried blood spots, venous blood was collected from the infant to analyze the total thyroxine (tT4) and TSH levels. Serum hormone TSH and tT4 were measured by an immune–enzymatic sandwich assay (ELISA) commercial kit (Roche Diagnostic, Mannheim, Germany). The cut-off venous tT4 concentration was 66 nmol/L (The National Center for Maternal and Child Health reference values, Ulaanbaatar, Mongolia). When venous TSH concentrations were > 10 mIU/L and the tT4 concentrations were <66 nmol/L, the infant was considered to have CH and was treated with L-thyroxine of 10 μg/kg every day. When the serum results (TSH and tT4) were within normal ranges, the infant was considered as false-positive. When TSH serum concentrations were between 12 and 16 mIU/L (gray zone) and the tT4 concentrations were normal, the patient was followed up every 15 days and considered as having subclinical hypothyroidism. A flowchart of the diagnosis of CH is shown in Figure 2.

### 2.3. Statistical Methods

All information was initially entered into a data collection sheet and then analyzed using Stata version 16 (StataCorp.2019. Stata Statistical Software: Release 16. College Station, TX: StataCorp LLC, College Station, TX, USA). Descriptive statistics, including frequencies, percentages, means, and standard deviations (SDs), median, and range were calculated to characterize the demographic and clinical data. The prevalence of CH was calculated using the total number of infants who were screened as a denominator and the total number of cases as the numerator. 

## 3. Results

From 2012 to 2020, a total of 23,002 newborns were screened and covered over 6.7% of the total neonatal population (approximately 40,000 births per year in Ulaanbaatar). The demographic characteristics of the population covered by the Mongolian National NBS program for CH and the results of the program are summarized in Table 1. The first year of the program covered three maternity hospitals from nine districts in Ulaanbaatar. The program expanded from three to six maternity hospitals (from a total of seven) of Ulaanbaatar in its eight years of existence, mainly in the central regions (Figure 1). The number of neonates screened for CH increased approximately two- to threefold in several years, particularly in 2017 and 2019, but it was not consistent during the study period (Table 1). However, when the total number of children born in Ulaanbaatar is considered, only 6.7% benefited from screening during the period of 2012 to 2020. In the study period, 87 children were suspected as having CH (total positive children at screening dried blood spot TSH in ≥ 20 mIU/L), representing an average of 0.4% of all of the samples analyzed. Inadequately collected blood samples, unsuitable for TSH determination, accounted for 3.0% on average by the guidance of IAEA and CDC [9,10]. Experience gained in blood sampling led to a decrease from the initial 5.3% to 1.6% more recently. 

All the positive children at screening (*n* = 87) were recalled for second screening, representing an average recall success rate of 0.4%, a value in accordance with international standards for TSH-based screenings [14]. From these infants, only 29 were sent for confirmation of CH, whereas 58 children were false-positive, representing an average of 66.7% of all positive children.

Finally, 24 of the 29 children were confirmed for primary CH, showing elevated TSH and decreased tT4 for those who were treated with T4 supplementation. We did not determine whether or not these patients had permanent primary hypothyroidism at the screening period. During the study period, the thyroid function of 13 infants (six infants were premature, five infants had intrauterine fetal growth retardation, and two infants had the mother’s antithyroid medication effect) normalized within six months of life, and thus, these infants were assumed to have transient CH; the remaining five had false-positive results. Thus, we identified 11 children with permanent CH, and we found that the prevalence of primary CH was 1 in 2091 in Ulaanbaatar, Mongolia.

Among the identified CH cases, four cases occurred in males, whereas seven cases were in females. The female-to-male ratio for CH cases was 1.8:1 in our study. The majority of the children were born by spontaneous vaginal delivery (72.7% of CH cases). The average birth weight was 3856.6 ± 235.5 g. The average birth height and head circumference were 48.2 ± 0.3 cm and 34.3 ± 1.1 cm, respectively. The median gestational age was 40; the range was five weeks. In addition, the average TSH level in dried blood spots was 33.8 ± 18.9 mIU/L. The median duration of treatment initiation was 15 days.

The majority of patients were asymptomatic (72.7%). Umbilical hernia and cold or mottled skin were the reported symptoms in our study (27.3%) (Table 2). Regarding the etiology of CH, a thyroid scan was performed in patients with permanent CH. Thyroid dysgenesis (hypoplasia and agenesis) was the most common etiology, with a total of nine cases (81.8% of cases) out of the 11 patients. Among them, cysts were found in two patients (18.2% of cases) with thyroid hypoplasia. Only two (18.2% of cases) patients had normal thyroid glands (Table 2). 

Finally, the lapse between the birth date and the initiation of L-thyroxine treatment in CH-positive children was lower than 15 days in 63.6% of cases and 15 to 30 days in 36.4% of children.

## 4. Discussion

In most developed countries, NBS is employed as a preventive and public health measure to identify and treat birth defects due to certain congenital conditions [14]. By government support, these developed countries/regions (Korea, Japan, Thailand, Singapore, Australia, New Zealand, and Hong Kong) have succeeded in reaching coverage rates of over 95% of their newborn population [4,14]. In Mongolia, a neonatal screening program for CH was started in 2000, the result of an international collaboration with IAEA, which was discontinued due to financial constraints in 2002.

In the present study, a total of 23,002 newborns (6.7% of annual births in Ulaanbaatar) were screened between 2012 and 2020 by measurement of the TSH from dried blood spots. Since this screening program was non-governmental, it received economic support from the Science and Technology Fund, Mongolia and Global Grand of Rotary International. The program reached a mean coverage of 6.7% of neonates born in participating maternity hospitals, increasing from the initial 12.4% to a maximum of 15.3%. However, our screening coverage was lower than levels of NBS coverage in the most economically developed countries, but many developing countries have only reached less than 1% of screening coverage due to other forms of infant screening competing with other health priorities, e.g., the control of infectious diseases, immunization, and malnutrition [15,16].

In our study, 24 neonates were confirmed as CH-positive. Of them, 11 neonates were permanent primary CH. This showed the largest national database on newborn screening to investigate an overall prevalence of CH of 1 per 2091 live births. The overall prevalence of primary CH in our present study was similar to the estimates of recent studies conducted in Ireland (1:2200) [17], China (1:2278) [12], the United States of America (1:2350) [18], Saudi Arabia (1:2470) [19], and Taiwan (1:1992) [3], However, similar or even higher results were found in Italy (1:1923) [20], Iran (1:1000) [21], Canada (1:1500) [22] and Srilanka (1:1652) [13].

In the first pilot study in Mongolia conducted by Erdenechimeg [8], the frequency of primary CH was slightly higher (1:1892) than in the present study. However, this pilot study was short term, and the number of participating children was low, which could have led to the conclusion concerning the high frequency of primary CH in Mongolia.

All countries with a coverage of at least 90% have fully integrated this screening program into their health delivery systems, including a payment scheme for an NBS fee. Payment is either covered by the government, insurance, or out-of-pocket expenses of the family [15]. However, to date, this health system has not been established in most developing countries, including Mongolia. In regard to the relatively high prevalence of CH in Mongolia due to not fully undetected risk factors and signs of CH often being subtle at birth, integration of universal screening programs among newborns into universal health coverage might be an appropriate public health policy option.

Most of the newborns are asymptomatic at birth, and only 5% to 10% of affected newborns had clinical signs and symptoms of CH [23]. More than four to six weeks after birth, children with severe CH may present with the symptoms and signs of CH, such as poor feeding, constipation, lethargy or excessive sleeping, failure to thrive, large anterior fontanelles, dry skin, prolonged jaundice, mottling, umbilical hernia, macroglossia, or coarse facial features [5]. In our study, most of the patients were asymptomatic (72.7%), which was considered slightly lower in comparison to the worldwide prevalence (90 to 95% are asymptomatic) [13]. In the majority of patients, CH is caused by an abnormal development of the thyroid gland (thyroid dysgenesis), which accounts for 80% to 85% of cases, and the remaining 10 to 15% of cases are caused by dyshormonogenesis [24]. In the present study, dysgenesis accounted for 81.8%, which was consistent with previous studies [23,25]. The average start of treatment was similar (average of 15.5 days) compared to other countries [26,27].

There were some limitations in the present study. First, due to the limited screening areas, particularly in the remote areas of Mongolia, we cannot determine the prevalence of CH in the whole country. Second, only 29 neonates out of 87 who tested positive for CH were retested for confirmation of CH, while 58 neonates were considered as false-positive results by our second screening. According to the guideline of the American Academy of Pediatrics, specimens collected in the first 24 to 48 h of life may lead to false-positive TSH elevations when using any screening test approach [28].

Therefore, we suspected that false-positive samples could occur due to early screening testing for CH (52.4% of cases were examined within 24 to 35 h) and due to early discharge from mothers who delivered by cesarean section (the majority of the children (61.5%) and children were discharged before 48 h of age). Lastly, we could not cover an adequate number of newborns in the screening program due to financial constraints. 

Consequently, the prevalence reported in our study may be underestimated, which means the true prevalence of CH in Mongolia could be higher than our estimation. Furthermore, a national NBS information system is needed to improve data, increasing the coverage of screening newborns, because not all areas have fully implemented this program. 

## 5. Conclusions

This is the first screening program to detect congenital hypothyroidism in the capital city of Ulaanbaatar, Mongolia. Over the course of eight years of NBS for CH, we covered only 6.7% of all those born in Ulaanbaatar; if we achieve 100% coverage of that, we can save around 19 to 20 children annually from mental and physical disabilities caused by CH. Furthermore, as the prevalence of CH seems to be increasing in the Mongolian population, future screening efforts should concentrate on expanding the coverage of the screening to the whole country.

## Figures and Tables

**Figure 1 IJNS-07-00029-f001:**
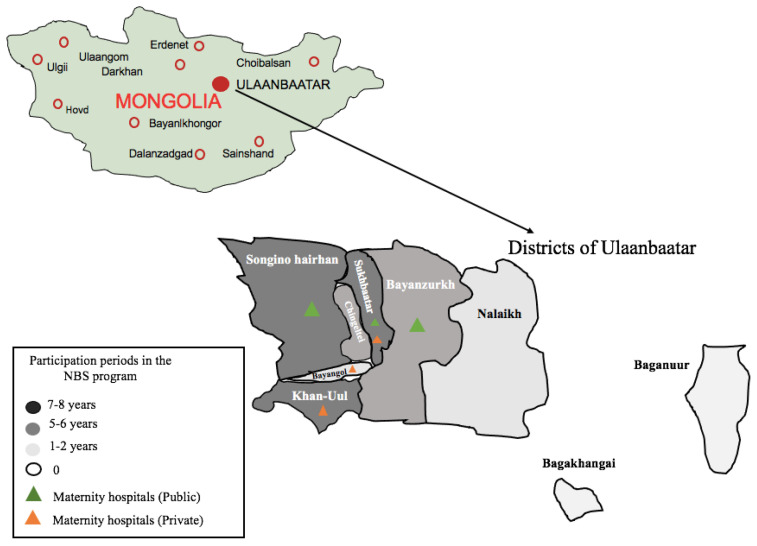
Participation periods (years) for the nine districts of Ulaanbaatar; six maternity hospitals are involved in the NBS program for CH. The map shows, in shades of light grey, dark grey, and black, the nine districts (out of nine in Ulaanbaatar); colored triangles show the six maternity hospitals (out of seven in Ulaanbaatar).

**Figure 2 IJNS-07-00029-f002:**
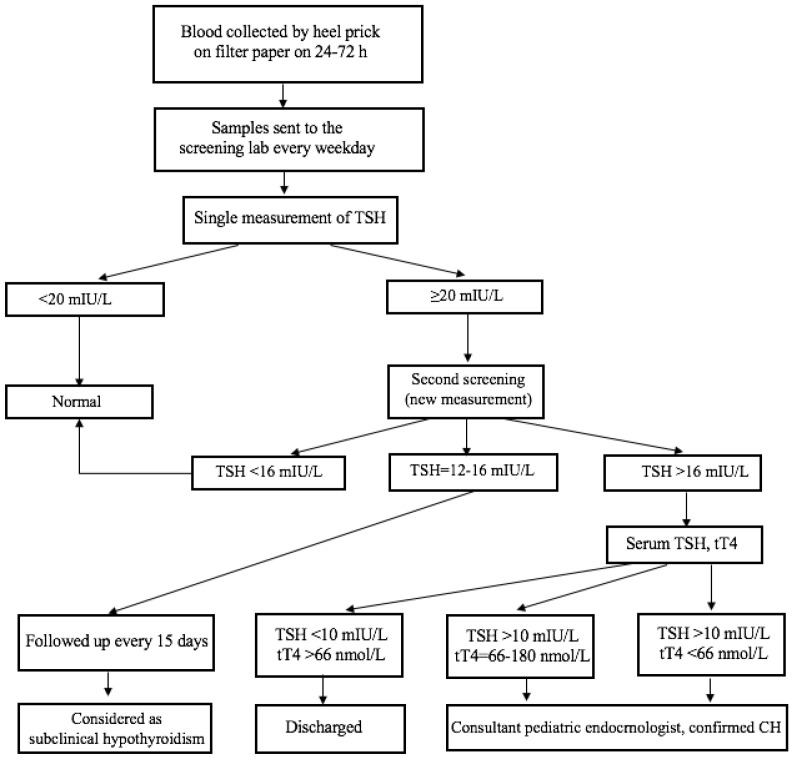
A flowchart of the diagnosis of CH; TSH-thyroid stimulating hormone, tT4-total thyroxine in serum.

**Table 1 IJNS-07-00029-t001:** Characteristics and results of the NBS for CH in Ulaanbaatar (2012 to 2020).

Year	2012 (between July and December)	2013	2014	2015	2016	2017	2018	2019	2020	Total/Average
**DEMOGRAPHY**										
Live births	18,607	41,187	41,786	41,731	40,687	38,404	40,703	41,206	39,779	344,090
Screened children	2304	1450	3095	686	2616	3269	461	6302	2819	23,002
Coverage (%)	12.4	3.5	7.4	1.6	6.4	8.5	1.1	15.3	7.1	6.7
**SCREENING**										
Positive children	11	5	22	2	6	4	1	25	11	87
Recall success rate (%)	0.5	0.3	0.7	0.3	0.2	0.1	0.2	0.4	0.4	0.4
False-positive (Second screening)	7	2	18	2	3	-	1	15	10	58
False-positive (%)	63.6	40	81.8	100	50	-	100	60	91	66.7
Inadequate samples ^1^	121	63	119	22	84	97	9	122	44	681
Inadequate samples (%)	5.3	4.3	3.8	3.2	3.2	3.0	2.0	1.9	1.6	3.0
**CONFIRMATION**										
Confirmed children with CH ^2^	4	3	4	-	3	4	-	10	1	29
Recall success rate (%) ^3^	36.4	60	18.2	-	50.0	100	-	40	9.1	33.3
Children with permanent CH	1	-	1	-	1	2	-	5	1	11
Children with transient CH	2	2	2	-	1	2	-	4	-	13
False-positive (Confirmation for CH)	1	1	1	-	1	-	-	1	-	5
False positive (%)	25	33.3	25	-	33.3	-	-	10	-	17.2

^1^ Inadequate sample for TSH determination due to faulty technique in dried blood samples collection, which were determined by the guidance of NBS from IAEA and CDC. ^2^ The number sent for confirmation of CH. ^3^ The percentage of children retested in serum for confirmation of CH.

**Table 2 IJNS-07-00029-t002:** Clinical manifestations of congenital hypothyroidism at the recall period.

Patient	Sex	Clinical Manifestations	Thyroid Ultrasound
1	F	Cold or mottled skin	Hypoplasia
2	F	Asymptomatic	Hypoplasia
3	F	Asymptomatic	Cysts with hypoplasia
4	M	Asymptomatic	Hypoplasia
5	F	Asymptomatic	Hypoplasia
6	F	Asymptomatic	Normal
7	F	Umbilical Hernia	Cysts with hypoplasia
8	M	Asymptomatic	Normal
9	M	Asymptomatic	Hypoplasia
10	F	Asymptomatic	Hypoplasia
11	M	Umbilical Hernia	Hypoplasia

## Data Availability

The data presented are available upon request from the corresponding author. The data are not publicly available because of privacy restrictions.

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
