# Peer review of "First Experiences with Newborn Screening for Congenital Hypothyroidism in Ulaanbaatar, Mongolia"

_2409-515X, 2021, doi:10.3390/ijns7020029_

Round 1
Reviewer 1 Report
This paper describes the first steps toward a regular neonatal screening programme for congenital hypothyroidism in Mongolia. Hence it fits into the scope of this journal and is worth publication.
However, there are a number of questions and suggestions for improvement.
- Number of births. It is stated that there are approximately 40000 births per year in Ulaanbaatar. Are all these births in the maternity hospitals? In Table 1 the number of participating maternity hospitals increase from 4 (to 3) to 6, but the number of live births increase from 18000 to 41000. This should be clarified.
- Number of screened infants. Table 1 states this number per year. Please explain the reasons for the large variations, e.g in 2014 a number of 3095 and in 2015 only686. A similar dip in 2018 and an enormous increase in 2019 to 6302. Why?
Why is the participation only 6%. Refusal of the parents? Financial constraints? Unwillingness of the health care providers? - Follow-up. It is stated “we did not determine whether or not these patients had permanent primary hypothyroidism”. Of course this should be done because this is fundamental for the success or failure of any screening programme. Why was this not done?
- The authors use 3 terms for the number of cases in respect to the test population, I.e incidence, frequency and prevalence. This is confusing. The correct term for this is “prevalence” (see e.g. https://www.physio-pedia.com/Epidemiology,_Prevalence_and_Incidence), because the detected infants already have the disorder, whereas “incidence” would indicate the detected children that acquire the disorder. Hence, “prevalence” should be used throughout this paper. The title could be changed into “First experiences with neonatal screening for Congenital hypothyroidism in Ulaanbaatar, Mongolia”.
- IJNS asks to use SI units. Please change µg/dl into nmol/L. The molecular weight of T4 is 777.
- A statistician should be consulted concerning the number of decimals in the numbers stated. Too many decimals would indicate a higher precision than is substantiated.
Author Response
Dear Reviewer,
We thank you for your careful reading of the manuscript and your helpful comments and suggestions. The comments and suggestions are valuable and very helpful for revising and improving our manuscript. We carefully examined your comments and tried to improve the paper in response, as outlined in a point-by-point manner on the following pages. Changes to the manuscript are indicated by tracked changes.
Comment 1:
Number of births. It is stated that there are approximately 40000 births per year in Ulaanbaatar. Are all these births in the maternity hospitals? In Table 1 the number of participating maternity hospitals increased from 4 (to 3) to 6, but the number of live births increased from 18000 to 41000. This should be clarified.
Response:
Thank you for your valuable comments. Approximately 40,000 births per year in maternity hospitals of Ulaanbaatar. Between 2012 and 2020, we have carried out this study. As shown in Table 1, the period from July to December 2012, 18607 live births were born in these maternity hospitals. This number indicated the number of live births born in Ulaanbaatar within a half-year period, because we had started our NBS in July 2012. At the beginning of our study, we trained neonatologists and nurses from public maternal hospitals. During the study period, private maternity hospitals were also connected. Therefore, the number of maternity hospitals increased from three to six.
Comment 2:
Number of screened infants. Table 1 states this number per year. Please explain the reasons for the large variations, e.g in 2014 a number of 3095 and in 2015 only686. A similar dip in 2018 and an enormous increase in 2019 to 6302. Why?
Why is the participation only 6%? Refusal of the parents? Financial constraints? Unwillingness of the health care providers?
Response:
We would like to thank the reviewer for pointing out the important issues. However, our study started in 2012, but we could not carry out this program constantly due to financial constraints. There are no other reasons such as parents' refusal, unwillingness of the health care providers and lack of team workers etc. We screened our neonates without any payment source from the government or family. Between 2012 and 2015, this study was partially supported by the Science and Technology Fund, Mongolia. From 2016 to 2020, our study was supported twice by the Rotary club of Niislel, Mongolia and Rotary club of Yongin, Korea, Rotary International Global Grand. During the study period, the screening program was temporarily discontinued due to financial sources, particularly in 2015 and 2018, when the project finished. Therefore, the number of screened children was too low in these years.
Comment 3:
Follow-up. It is stated, “we did not determine whether or not these patients had permanent primary hypothyroidism”. Of course, this should be done because this is fundamental for the success or failure of any screening program. Why was this not done?
Response:
We appreciate your comments. We screened a total of 23002 newborns and confirmed 24 children with primary CH (including permanent and transient). At the screening period, we did not determine whether or not these patients had permanent primary hypothyroidism. During the follow-up period, the thyroid function of 13 infants normalized within six months of life, and thus, these infants were assumed to have transient CH. Therefore, 11 infants had permanent CH. The prevalence of primary CH was 1 in 2091.
Comment 4:
The authors use 3 terms for the number of cases in respect to the test population, I.e incidence, frequency and prevalence. This is confusing. The correct term for this is “prevalence” (see e.g. https://www.physio-pedia.com/Epidemiology,_Prevalence_and_Incidence), because the detected infants already have the disorder, whereas “incidence” would indicate the detected children that acquire the disorder. Hence, “prevalence” should be used throughout this paper. The title could be changed into “First experiences with neonatal screening for Congenital hypothyroidism in Ulaanbaatar, Mongolia”.
Response:
We have modified the term “prevalence” throughout the manuscript. We have also revised the title into “First experiences with newborn screening for Congenital hypothyroidism in Ulaanbaatar, Mongolia”.
Comment 5:
IJNS asks to use SI units. Please change µg/dl into nmol/L. The molecular weight of T4 is 777.
Response: Thank you very much for your kind suggestion. We have modified µg/dl into nmol/L in the revised manuscript.
Comment 6:
A statistician should be consulted concerning the number of decimals in the numbers stated. Too many decimals would indicate a higher precision than is substantiated.
Response: We rounded 2 decimal places to 1 decimal place throughout the revised manuscript (0.38% to 0.4%etc.).
Sincerely,
Erdenetuya Ganbaatar, MD., PhD
Department of Pediatrics, School of Medicine, Mongolian National University of Medical Sciences, Ulaanbaatar, Mongolia
Tel: +976-9917-5770
E-mail: erdenetuya@mnums.edu.mn
Reviewer 2 Report
This paper describes the pilot screening program to detect congenital hypothyroidism in infants born in Ulaanbaatar, Mongolia. Whilst is general the paper is well written there are a few suggestions to improve and or clarify the content.
Overall:
- The references seem somewhat inappropriate. There are many articles that could be referenced for the generalisations eg the first sentence of the introduction could have many references.
- A number of references are quite old when discussing current trends eg incidence of 1 in 3000 to 4000 live births worldwide. This is not consistent with the paper on NBS worldwide in 2015.
Abstract:
- The first sentence could be restructured: Congenital hypothyroidism (CH) is among the most common conditions leading to intellectual disability which can be prevented by early detection through newborn screening.
- Hypoplasia is indicated in 9 cases in Table 3. Two with cysts with hypoplasia. Should the total therefore be 9 rather than 7?
Introduction:
- CH among Asian infants is higher than among White and Black infants. It is suggested that White and Black be changed to "other ethnicities"
- The references should be reviewed eg ref 3 is increase in New York, ref 7 refers to NGS, ref 8 is for Oman. Ref 9 used to indicate assay methods is refers to Ireland. Ref 10 is used to cite increased awareness of the frequency.
Materials and Methods :
- Patient population: Paragraph should be restructured to keep like thoughts together: From 2012 to 2020, a total of 23,002 neonates were screened in Ulaanbaatar for CH. During this period a total of nine districts joined the screening program, involving a maximum of 6 maternity hospitals (Fig 1). The suspected neonates were recalled to confirm CH. CH was diagnosed with a report of thyroid ultrasound and the results of laboratory tests. Ultrasonography of the thyroid gland was performed during the first month of life.
- Figure 1: Could indicate where the maternity hospitals are located. The key should include "years" eg 7-8 years etc. The title needs correction.
Neonatal Screening for CH
- There seems some confusion whether measurements are free or total thyroxine.
- venous blood was collected - was this analysed as serum or dried blood spot? Note the concentration is different depending on the matrix
- There is a typo >5.1ug/dl should be <5.1 for infants considered as CH
Figure 2:
- As is common practice, is the single TSH measurement repeated on the same blood sample? If so, this box should be included
- Is the thyroxine free or total?
- Note the difference in values between serum and blood in the title or a footnote
Table 1:
- The header row should state Total/average
- The inclusion of inadequate samples does not add relevant information
- False positive is defined as any one where a further sample was requested not just those who also had diagnostic samples requested.
- 'Confirmed children with CH' is the number sent for confirmation
- The data for 2014 indicates 4 sent for confirmation with 1 CH, 1 transient and 1 FP. What was the other one?
Tables 2 and 3:
- States 4 Male and 7 Female - however in table 3 there are 3 male and 8 female. Please correct
- The gestational age, mean +/-SD is 41.5+/-1.3. It would be more appropriate to indicate median and range as the total is only 11.
- DBS TSH mean+/- SD is 33.8+/-8.9. This seems very low for TSH with this methodology. Please check
Results:
- The majority of patients (8/11) were asymptomatic. If only 1 had cold or mottled skin and 2 umbilical hernia it is a stretch to indicate these as most reported symptoms
- Table 3 indicates 9 patients with hypoplasia yet in the text it is stated as 7. Please check and correct
Discussion
- See comment above on references cited.
- In Mongolia, a neonatal screening program for CH was started in 2000 - what happened to that study? Was it completed in 2002?
- The number of neonates increased approximately 2-3 fold in several years - this is not evident in the table
- Discussion of comparative coverage should be more up to date - reference 14 is from 2006, others 2007 or 2008
- Please explain how laboratory methods may be associated with incidence of CH?
Author Response
Dear Reviewer,
We thank you for your careful reading of the manuscript and helpful comments and suggestions. The comments and suggestions are valuable and very helpful for revising and improving our manuscript. We carefully examined your comments and tried to improve the paper in response, as outlined in a point-by-point manner on the following pages. Changes to the manuscript are indicated by tracked changes in the revised manuscript.
Overall:
Comment 1:
The references seem somewhat inappropriate. There are many articles that could be referenced for the generalizations e.g. the first sentence of the introduction could have many references.
Response:
Thank you very much for your valuable comments. We re-cited the references in the revised manuscript.
Comment 2:
A number of references are quite old when discussing current trends e.g. incidence of 1 in 3000 to 4000 live births worldwide. This is not consistent with the paper on NBS worldwide in 2015.
Response:
Thank you very much for your valuable comments. We modified some references and re-cited them in the revised manuscript.
Abstract:
Comment 3:
The first sentence could be restructured: Congenital hypothyroidism (CH) is among the most common conditions leading to intellectual disability which can be prevented by early detection through newborn screening.
Response:
According to your kind suggestions, we have re-structured the first sentence of the abstract.
Comment 4:
Hypoplasia is indicated in 9 cases in Table 3. Two with cysts with hypoplasia. Should the total therefore be 9 rather than 7?
Response:
Thank you very much for pointing this out. We have corrected 7 to 9 in the revised manuscript.
Introduction:
Comment 5: CH among Asian infants is higher than among White and Black infants. It is suggested that White and Black be changed to "other ethnicities
Response: According to your kind suggestions, we re-structured this sentence in the introduction.
Comment 6:
The references should be reviewed eg ref 3 is increase in New York, ref 7 refers to NGS, ref 8 is for Oman. Ref 9 used to indicate assay methods refers to Ireland. Ref 10 is used to cite increased awareness of the frequency.
Response:
We have reviewed all references in the revised manuscript.
Materials and Methods:
Comment 7:
Patient population: Paragraph should be restructured to keep like thoughts together: From 2012 to 2020, a total of 23,002 neonates were screened in Ulaanbaatar for CH. During this period a total of nine districts joined the screening program, involving a maximum of 6 maternity hospitals (Fig 1). The suspected neonates were recalled to confirm CH. CH was diagnosed with a report of thyroid ultrasound and the results of laboratory tests. Ultrasonography of the thyroid gland was performed during the first month of life.
Response:
We would like to thank for your kind suggestions. We re-structured this paragraph in the material and methods.
Comment 8:
Figure 1: Could indicate where the maternity hospitals are located. The key should include "years" e.g. 7-8 years etc. The title needs correction.
Response: Thank you very much for your valuable comments. We have indicated where the maternity hospitals are located, which are shown in colors of triangular forms in Figure 1. We have also corrected “years'' either in the figure or title.
Neonatal Screening for CH
Comment 9:
There seems some confusion whether measurements are free or total thyroxine.
venous blood was collected - was this analyzed as serum or dried blood spot? Note the concentration is different depending on the matrix
Response: We appreciate your comments. Venous blood was collected from the infant to analyze the total thyroxine (tT4) and TSH levels in serum, not dried blood spots. Serum hormone TSH and tT4 were measured by an immune–enzymatic sandwich assay (ELISA).
Comment 10:
There is a typo >5.1ug/dl should be <5.1 for infants considered as CH
Response: Thank you very much for pointing this out. We have corrected >5.1μg/dl to <5.1 μg/dl. According to the comments from another Reviewer, we have modified μg/dl into nmol/L (page 3; line 110 and 113).
Figure 2:
Comment 11: As is common practice, is the single TSH measurement repeated on the same blood sample? If so, this box should be included
Response: Thank you for your comments. TSH values ≥20 mIU/L on the filter paper were considered positive for CH, and a new dried blood specimen was requested for the second screening. We have not checked the same blood sample for the second screening. We revised Figure 2.
Comment 12:
Is the thyroxine free or total?
Response:
We appreciate your pointing this out. In this study, we measured the total thyroxine. Thus, we have corrected it throughout the revised manuscript.
Comment 13:
Note the difference in values between serum and blood in the title or a footnote
Response: We appreciate your kind suggestions. Serum hormone TSH and total T4 were measured by an immune–enzymatic sandwich assay (ELISA). We have noted it in the title of the Figure 2.
Table 1:
Comment 14: The header row should state Total/average
Response: According to your kind suggestions, we have modified total/average in Table 1.
Comment 15: The inclusion of inadequate samples does not add relevant information
Response:
We appreciate your kind suggestions; we have noted the relevant information of the inclusion of inadequate samples in thefootnote of Table 1. Inadequately collected blood samples or invalid blood specimens for TSH determination were determined by the guidance from IAEA and CDC. We added this information in materials and methods as well (page 3; line 93-96).
Comment 16:
False positive is defined as anyone where a further sample was requested not just those who also had diagnostic samples requested.
Response:
Thank you for pointing this out. We have corrected it in Table 1.
Comment 17:
Confirmed children with CH' is the number sent for confirmation
Response:
This is the number sent for confirmation. We noted it in the footnote of Table 1.
Comment 18: The data for 2014 indicates 4 sent for confirmation with 1 CH, 1 transient and 1 FP. What was the other one?
Response: Thank you for your comments. We mistyped “2” to “1” in this table. We have corrected the number in table 1.
Tables 2 and 3:
Comment 19: States 4 Male and 7 Female - however in table 3 there are 3 males and 8 females. Please correct
Response: Thank you very much for pointing this out. We have reviewed and corrected it in table 2 (patient number 4 was male, not female).
Comment 20: The gestational age, mean +/-SD is 41.5+/-1.3. It would be more appropriate to indicate median and range as the total is only 11.
Response: According to your kind suggestions, we have modified and mentioned it in the text. Because we omitted Table 2 due to the comments from another Reviewer (page 7; line 175).
Comment 21: DBS TSH mean+/- SD is 33.8+/-8.9. This seems very low for TSH with this methodology. Please check
Response: Thank you for your comments. We have mistyped this number and corrected 33.8±8.9 to 33.8±18.9 in the revised manuscript (page 7; line 176). However, our TSH mean has not been changed after double checking our data analysis.
Results:
Comment 22: The majority of patients (8/11) were asymptomatic. If only 1 had cold or mottled skin and 2 umbilical hernia it is a stretch to indicate these as most reported symptoms
Response: We appreciate your kind comments. We re-structured the sentence (page 7; line 183).
Comment 23:
Table 3 indicates 9 patients with hypoplasia yet in the text it is stated as 7. Please check and correct
Response: We have corrected it in the text (page 7; line 185)
Discussion
Comment 24: See comment above on references cited.
Response: Thank you for pointing this out. We re-cited the references.
Comment 25:
In Mongolia, a neonatal screening program for CH was started in 2000 - what happened to that study? Was it completed in 2002?
Response: In Mongolia, a neonatal screening program for CH was started in 2000, the result of an international collaboration with IAEA, which was discontinued due to financial constraints since 2002. Between 2012 and 2015, this study was partially supported by the Science and Technology Fund, Mongolia. From 2016 to 2020, our study was supported twice by the Rotary club of Niislel, Mongolia and Rotary club of Yongin, Korea, Rotary International Global Grand.
Comment 26:
The number of neonates increased approximately 2-3-fold in several years - this is not evident in the table
Response:
This sentence described the number of screened newborns for CH during this study. However, we have started to screen 2304 newborns in 2012 whereas 6302 newborns were screened by us in 2019. During the study period, the screening program was temporarily discontinued due to financial sources, particularly in 2015 and 2018, when the project finished. Therefore, the number of screened children was too low in these years.
Comment 27:
Discussion of comparative coverage should be more up to date - reference 14 is from 2006, others 2007 or 2008
Response:
Thank you very much for your valuable comments. We modified some references and re-cited them in the discussion.
Comment 28:
Please explain how laboratory methods may be associated with incidence of CH?
Response: A previous study reported that although the use of different laboratory methods and screening practices by NBS laboratories affected the incidence rate of CH, after adjusting for screening methodologies and parameters, an increasing incidence rate still persisted during the decade studied. Thus, there seem to be additional unknown factors that contributed to the reported increase in incidence rate (Hertzberg V, Mei J, Therrell BL. Effect of laboratory practices on the incidence rate of congenital hypothyroidism. Pediatrics. 2010 May;125 Suppl 2:S48-53. doi: 10.1542/peds.2009-1975E. PMID: 20435717). Moreover, lowering of the TSH cutoff was the most important factor contributing to the increase of CH incidence in Italy(Olivieri A, Fazzini C, Medda E; Italian Study Group for Congenital Hypothyroidism. Multiple factors influencing the incidence of congenital hypothyroidism detected by neonatal screening. Horm Res Paediatr. 2015;83(2):86-93. doi: 10.1159/000369394). 
Sincerely,
Erdenetuya Ganbaatar, MD., PhD
Department of Pediatrics, School of Medicine, Mongolian National University of Medical Sciences, Ulaanbaatar, Mongolia
Tel: +976-9917-5770
E-mail: erdenetuya@mnums.edu.mn
Reviewer 3 Report
The reviewed article presents the results of the incidence of CH detected by NS in Ulanbaatar Mongolia in the years 2012 - 2020 by determining TSH (Delfia assay) the first 24-72 hours after birth.
- / This is a pilot study from obstetrics in the capital Ulaanbaatar, which aims to present these first results with neonatal CH screening in Mongolia. From this point of view, the Introduction is unnecessarily extensive. I recommend that it be substantially reduced. Also, the goal of looking for potential environmental impacts on the incidence of CH is inappropriate for incomplete NS (Discussion and Abstract)
- / In the text on further diagnostic steps on page 3, line 13, there are discrepancies: If in the second screen the TSH level is above 16 mIU / L in venous blood, free thyroxine (FT4) TSH is determined. Subsequently, TSH and total T4 (TT4) were determined by ELISA. The FT4 cutoff is reported to be 5.1 ug / dL and it is further stated that the criterion for the diagnosis of CH is TSH above 10 mIU / L and the FT4 above (not below !!) 5.1 ug / dL. This is followed by treatment with L-thyroxine. The data are in conflict with the data in Figure 2 (where Figure 1 is?), Where the diagnosis of CH determines TSH and total T4 (TT4) with the interface shown as in the text. So, what is reality? FT4 or TT4? The normal range fT4 is reported as 0.27 - 6.7 mU / L, the TT4 range 11-23 pmol / L.
- / The results in Table 1 summarize: 3.1./ In the first step of the NS, only 6.7% of births from the tested Ulaanbaatar maternity wards were detected, which is insufficient coverage of the selected cohort of the population. 87 cases (which is a recall rate of 0.38% - not 0.36%) caught in the first step were subjected to a second tier examination. This is not mentioned. Is it venous TSH and TT4, or fT4? 3.2./ Out of 87 suspected cases of CH, 29 children were confirmed as CH, which represents 33.33% It is not clear why the table shows the recall success rate of the total number of NS examined children (23002). The definitive number of cases of CH is 11. 13 children had transient hypothyroidism. The status of the remaining 5 children (11 + 13 = 24, 29 - 24 = 5?) Is not listed - in the table as "false positive" - ​​in which diagnostic step? 3.3./ Table 1 lists up to 681 inadequate samples (2.96%) not explained how they were included / discarded.
- Summary of point 3. / I recommend deleting the confusing and not entirely clear Table 1 and presenting the essential data from it in the relevant window of Graph 2 in absolute number and percentage. Percentage not of the total number, but of the corresponding diagnostic step. 4. / Data on the clinical characteristics of patients with CH are in agreement with generally known data. I recommend omitting Table 2, just a brief mention in the text. 5. / I recommend reducing the discussion in proportion to your own work. This is not an overview work on CH. 6. / On page 7 5, the line from the bottom should be quoted 11 instead of 9 Conclusion: I recommend to fundamentally rework the work according to the objections,
Author Response
Dear Reviewer,
We thank you for your careful reading of the manuscript and your helpful comments and suggestions. The comments and suggestions are valuable and very helpful for revising and improving our manuscript. We carefully examined your comments and tried to improve the paper in response, as outlined in a point-by-point manner on the following pages. Changes to the manuscript are indicated by tracked changes in the revised manuscript.
Comment 1:
This is a pilot study from obstetrics in the capital Ulaanbaatar, which aims to present these first results with neonatal CH screening in Mongolia. From this point of view, the Introduction is unnecessarily extensive. I recommend that it be substantially reduced. Also, the goal of looking for potential environmental impacts on the incidence of CH is inappropriate for incomplete NS (Discussion and Abstract).
Response:
We appreciate your kind suggestions. We have reduced the introduction and restructured the sentences in the discussion and abstract.
Comment 2:
In the text on further diagnostic steps on page 3, line 13, there are discrepancies: If in the second screen the TSH level is above 16 mIU / L in venous blood, free thyroxine (FT4) TSH is determined. Subsequently, TSH and total T4 (TT4) were determined by ELISA. The FT4 cutoff is reported to be 5.1 ug / dL and it is further stated that the criterion for the diagnosis of CH is TSH above 10 mIU / L and the FT4 above (not below !!) 5.1 ug / dL. This is followed by treatment with L-thyroxine. The data are in conflict with the data in Figure 2 (where Figure 1 is?), Where the diagnosis of CH determines TSH and total T4 (TT4) with the interface shown as in the text. So, what is reality? FT4 or TT4? The normal range fT4 is reported as 0.27 - 6.7 mU / L, the TT4 range 11-23 pmol / L.
Response:
We would like to thank the reviewer for pointing out the important comments. We have corrected >5.1μg/dl to <5.1 μg/dl. According to the comments from another Reviewer, we have modified μg/dl into nmol/L (page 3; line 101 and 103). In this study, we measured the total thyroxine (tT4), not free thyroxine. We mistyped it in the text. Thus, we have corrected free T4 into total T4 throughout the manuscript. As shown in page 3; line 106-114, we stated the diagnosis of CH determines TSH and total T4 (tT4) with the interface. Figure 1 showed the participation periods for the nine districts of Ulaanbaatar, which also showed six maternity hospitals were involved in the NBS program for CH.
Comment 3:
The results in Table 1 summarize: 3.1./ In the first step of the NS, only 6.7% of births from the tested Ulaanbaatar maternity wards were detected, which is insufficient coverage of the selected cohort of the population. 87 cases (which is a recall rate of 0.38% - not 0.36%) caught in the first step were subjected to a second-tier examination. This is not mentioned. Is it venous TSH and TT4, or fT4? 3.2./ Out of 87 suspected cases of CH, 29 children were confirmed as CH, which represents 33.33% It is not clear why the table shows the recall success rate of the total number of NS examined children (23002). The definitive number of cases of CH is 11. 13 children had transient hypothyroidism. The status of the remaining 5 children (11 + 13 = 24, 29 - 24 = 5?) Is not listed - in the table as "false positive" - ​​in which diagnostic step? 3.3./ Table 1 lists up to 681 inadequate samples (2.96%) not explained how they were included / discarded.
Response:
Thank you for your valuable comments. Over the course of 8 years of NBS for CH, we covered only 6.7% of all those born in Ulaanbaatar. Because we could not cover an adequate number of newborns in the screening program due to financial constraints. We reviewed the number and modified 0.36% to 0.38% in the revised manuscript, which mentioned it in the text (page 6, line 153). According to the comments from another Reviewer, we rounded 2 decimal places to one decimal place throughout the manuscript (0.38% to 0.4%). We corrected the recall rate for the confirmation test (0.13% to 33.3% in table 1). During the study period, out of 29 suspected cases for primary CH, 11 children were confirmed for permanent CH and 13 infants were transient CH. The remaining 5 infants had a false-positive result (17.2%), which showed in table 1. Moreover, all inadequately collected blood samples or invalid blood specimens for TSH determination were determined by the guidance from IAEA and CDC, these samples were excluded and recollected by trained nurses at the maternity hospitals (page 3; line 93-37).
Comment 4:
Summary of point 3. / I recommend deleting the confusing and not entirely clear Table 1 and presenting the essential data from it in the relevant window of Graph 2 in absolute number and percentage. Percentage not of the total number, but of the corresponding diagnostic step. 4. / Data on the clinical characteristics of patients with CH are in agreement with generally known data. I recommend omitting Table 2, just a brief mention in the text. 5. / I recommend reducing the discussion in proportion to your own work. This is not an overview work on CH. 6. / On page 7 5, the line from the bottom should be quoted 11 instead of 9 Conclusion: I recommend to fundamentally rework the work according to the objections
Response:
Thank you very much for your valuable comments. We have corrected Table 1. Moreover, we omitted Table 2 and mentioned the data in the text (page 6, line 172-179). All references were re-reviewed and re-cited in the revised manuscript. According to your kind suggestions, we reduced and restructured the discussion. Based on the goal, we reworked all parts of the manuscript.
Sincerely,
Erdenetuya Ganbaatar, MD., PhD
Department of Pediatrics, School of Medicine, Mongolian National University of Medical Sciences, Ulaanbaatar, Mongolia
Tel: +976-9917-5770
E-mail: erdenetuya@mnums.edu.mn
Round 2
Reviewer 3 Report
After corrections the manuscript is optimal for publication